# Conservation of the Bird Cherry (*Padus* Mill.) Germplasm by Cold Storage and Cryopreservation of Winter Cuttings

**DOI:** 10.3390/biology12081071

**Published:** 2023-07-30

**Authors:** Vladimir Verzhuk, Sergey Murashev, Liubov Novikova, Stepan Kiru, Svetlana Orlova

**Affiliations:** 1N.I. Vavilov Institute of Plant Genetic Resources, 190000 Saint-Petersburg, Russia; vverzhuk@mail.ru (V.V.); l.novikova@vir.nw.ru (L.N.); s.orlova@vir.nw.ru (S.O.); 2Department of Storage Technology and Processing of Agricultural Products, Saint-Petersburg State Agrarian University, 196605 Saint-Petersburg, Russia; s.murashev@mail.ru; 3Department of Crop Production named I.A. Stebut, Saint-Petersburg State Agrarian University, 196605 Saint-Petersburg, Russia

**Keywords:** liquid nitrogen vapor storage, plant germplasm cryopreservation, fruit tree conservation, genebank, cold storage, fruit biochemistry, ascorbic acid

## Abstract

**Simple Summary:**

This is the first study describing successful recovery of winter cuttings from five bird cherry varieties of different genetic origin after six months of cryopreservation in liquid nitrogen vapor (−183–−185 °C). This study also included analysis of morphometric data collected for plants developed from cryopreserved cuttings, and biochemical analysis of fruits produced by plants after cryopreservation in the field during three consecutive years. The viability of cuttings recovered after six months of cryopreservation varied from 43 to 50% which exceeded the internationally accepted genebank viability standard (40%). Cryopreservation had little to no impact on the morphological parameters of the developed plants and no influence on the biochemical composition of the fruits. All parameters measured for plants after cryopreservation were comparable to those recorded after cold storage at −5 °C, which implies suitability of these storage methods for long- and mid-term conservation, respectively, of the bird cherry genetic collection.

**Abstract:**

Conservation at cryogenic temperatures, usually in liquid nitrogen (LN) or in its vapor, is the only reliable method for the long-term ex situ conservation of fruit and berry crops with vegetative reproduction. In this study, five bird cherry (*Padus* Mill.) varieties of different genetic origin from the bird cherry genebank at the N.I. Vavilov All-Russian Institute of Plant Genetic Resources (VIR, Russia) were tested for their response to cryopreservation in LN vapor (−183–−185 °C). The response included viability under laboratory and field conditions, morphological assessment of the developed plants and biochemical analysis of fruits produced during three consecutive years by plants developed from cryopreserved cuttings. All parameters were compared to those recorded after cold storage of cuttings (−5 °C), a routine mid-term conservation method currently used at the VIR genebank. The initial viability of winter cuttings varied from 86.7% to 93.3%. Six-month cold storage and cryopreservation reduced viability to 53.3–86.7% and 43.3–60.0%, respectively, which was above the 40% viability threshold in all varieties tested. Cuttings after cold storage showed better viability when recovered in the laboratory (80% mean viability) than in the field (58% mean viability); viability of cryopreserved cuttings was not affected by recovery conditions. The results of a two-way analysis of covariance suggested that storage and recovery conditions had the most significant effect on viability (*p* < 0.0001), while the effects of genotype (*p* = 0.062) and factor interactions (*p* = 0.921) were minor. Cryopreservation had little or no influence on morphological parameters of the plants recovered in the field, including plant height, number of shoots, internodes and roots, and root length. Similarly, no effect of cryopreservation was recorded on dry matter content, total sugar content and ascorbic acid concentration in fruits produced by plants developed from the cryopreserved cuttings. These results suggest that cryopreservation in LN vapor is a reliable method for conservation of the bird cherry genetic collection and is worth testing with a broader variety of genotypes.

## 1. Introduction

Cryobiology is the branch of biology that studies effects of low temperatures on living organisms. F. Simon, a British physicist, noted that “… this is the field where human has far surpassed nature itself” [1]. Implementation of cryogenic temperatures opened new directions to investigating biophysical properties of living cells and tissues and stimulated the development and application of new research methods and technologies [2]. From the practical viewpoint, the greatest advantage of storage at ultra-low temperatures is the significant deceleration or even complete stop of metabolic processes in plant or animal tissues [3]. It is acknowledged that cryogenically stored material remains genetically stable and thus unsusceptible to genetic changes that may occur in living organisms conserved under ambient conditions [4]. Plant biodiversity can be conserved in situ, i.e., in natural environments, such as nature reserves or fields, or ex situ—in collector’s gardens, nurseries or genebanks. For most of the cultivated plants, germplasm is stored in the form of seeds in genebanks at positive (+4–+5 °C) or negative (−10–−18 °C) temperatures, thus ensuring high viability of seeds even after decades of storage [5,6,7]. However, this method is not applicable for plants with vegetative propagation, because seed propagation does not ensure preservation of agriculturally important traits in the offspring. The most effective method of ex situ storage of fruit and berry crops with vegetative propagation is cryopreservation in liquid nitrogen (LN, −196 °C) or its vapor phase (−183–−185 °C) [8,9,10,11]. This method was successfully applied for the long-term storage of meristem and pollen of the blackcurrant, apple, sweet cherry, and plum [12,13,14,15,16,17,18,19,20]. Cryopreservation using winter-dormant vegetative buds was implemented to back-up field collections of *Malus* in the USA [21,22], Canada [23], and Germany [24]. The important advantages that cryopreservation of dormant buds can offer, compared to other storage methods, include no impact on cultivar genetic integrity, significantly lower maintenance costs, smaller areas for sample storage, and an indefinitely long storage period [25]. Despite the high initial costs and a complex procedure of putting samples into cryogenic storage, cryopreservation is the only method that ensures stability of the main cytogenetic characteristics of plant materials over the decades of storage—an advantage that cannot be achieved using any other method, including cold storage [26,27].

The common European bird cherry (*Padus avium* Mill., synonym of *Prunus padus* L.) is widely spread through the whole Russian territory, from its western to eastern borders. Another bird cherry species, *P. virginiana* L., has been introduced to Russia from North America. The bird cherry is the most frost-resistant among all stone fruit crops and is widely grown for fruit production and greenspace extension in central and northern regions of the country. The tree is undemanding and easy to cultivate. Fruits, leaves, bark, and flowers of the bird cherry have high pharmacological value and have been used for centuries in folk medicine. The fruits are used in the human diet, mostly in the regions of West Siberia. Improved varieties of bird cherry with high nutritional and decorative values were developed in the Central Siberian Botanical Garden after crossbreeding with the chokecherry. At present, intensive research and breeding of the bird cherry aims to develop new cultivars with higher fruit quality; therefore, chemical composition of fruits including sugars and ascorbic acid content is of primary importance [28,29,30].

Field collections of diverse varieties of bird cherry are maintained at the research stations of the All-Russian Institute of Plant Genetic Resources (VIR) (St. Petersburg, Russia). Bird cherry is the only stone fruit plant that can be propagated by ligneous cuttings in the spring. Therefore, cold storage of winter scions with dormant buds at −5 °C and 60% air humidity is currently used as the main mid-term storage method to back up field collections of bird cherry.

This research aimed to explore the suitability of cryopreservation in LN vapor for conservation of bird cherry cuttings using five varieties of different genetic origin from VIR collection. The viability of cuttings after cryogenic storage, morphometric analysis of the developed plants, and biochemical analysis of fruits collected from plants grown after cryopreservation were measured and compared to those recorded after conventional cold storage.

## 2. Materials and Methods

### 2.1. Plant Material

Five varieties of the bird cherry (*Padus* Mill.) of different genetic origins were selected from the bird cherry genebank of the Research Station Pushkin and Pavlovsk Laboratory of the N.I. Vavilov, All-Russian Institute of Plant Genetic Resources (VIR) (30 km from St. Petersburg, Russia) (Table 1). Cryopreservation studies were carried out at the Laboratory for Long-Term Storage of Plant Genetic Resources of VIR (St. Petersburg, Russia). Biochemical composition of bird cherry fruits was analyzed at the Biochemistry Laboratory of St. Petersburg State Agrarian University (St. Petersburg, Russia).

### 2.2. Cold Storage and Cryopreservation

The experiments were conducted over three consecutive years (2013–2015). In December, dormant one-year-old scions, 25–30 cm long, were collected from trees in the Research Station and transported to the Laboratory for Long-Term Storage of Plant Genetic Resources. In the laboratory, the scions were divided into 6–8 cm long segments (cuttings), each having 2 or 3 buds. The cuttings were randomly divided into three groups that were used for control without treatments (baseline viability assessment), cold storage, and cryopreservation. An equal number of cuttings (200 pieces) was used for each treatment.

To determine the initial (baseline) viability, cuttings collected from the field were partially placed in tap water and kept at 21 °C and a 16/8 h light/dark regime, illumination of 5000–7000 lux was provided by cool fluorescent lamps in a climate control chamber (hereafter designated as “laboratory conditions”). The viability was assessed after 160 days as a percentage of cuttings that produced normal looking new stems, leaves, and roots.

The cuttings of the second group (cold storage treatment) were stored during 6 months at −5 °C, 16/8 h light/dark regime, air humidity 60%; then their viability was determined under laboratory conditions (described above) and in the field [31,32,33]. To assess viability in the field, immediately after cold storage the cuttings were transferred to the Research Station and planted in soil under ambient conditions. The viability was assessed after 160 days as a percentage of the cuttings that produced normal looking new stems, leaves, and roots.

The cuttings of the third group intended for cryopreservation were dried in a thermostat at −4~−5 °C to a water content of 28–32% based on the fresh weight (determined gravimetrically). Cryopreservation was performed using the method suggested by Forsline et al. [21] for cryopreservation of apples. Dried cuttings were placed into laminated bags, 10 × 15 cm, (10 cuttings per bag) and frozen in the Sanyo Medical Freezer U442 (T) (Japan) using a two-step technique. In the first step, cuttings were frozen at a rate of −1~−2 °C per minute to −28~−32 °C. Then the freezing rate was increased to −3~−4 °C per minute until reaching a terminate temperature of –50 °C. Then the bags were quickly transferred to LN vapor (−183~−185 °C) for storage. After 6 months of cryogenic storage, bags with cuttings were rewarmed in a water bath at +18~+20 °C. Cuttings were withdrawn from the bag and their viability was assessed in the laboratory or in the field as described above.

Cold storage is currently a routine conservation method in the VIR genebank; therefore, cuttings after cold storage were used for comparison of morphometric parameters and fruit composition after cryopreservation. 

### 2.3. Comparison of the Morphometric Parameters

Morphometric parameters such as plant height, the number of shoots and internodes, and the number and length of roots were assessed in 15–20 randomly selected rooted plants from cold storage and cryopreservation treatments recovered in the laboratory and in the field 160 days after planting. Morphological observations were performed in compliance with the Program and Methods for Research on Varieties of Fruit, Berry and Nut Crops [34].

### 2.4. Fruits Biochemistry

Mature fruits (total weight 500–1000 g) were collected for three years (2017 to 2019) from plants grown in the field after cold and cryogenic storage (5 plants per storage conditions per variety). Dry matter content, total sugar content, and the content of ascorbic acid were measured as the key parameters of fruit quality [35]. The dry matter content was determined by drying fruits at 105 °C until a constant weight was achieved. The concentration of ascorbic acid in raw fruits was determined by titration with Tillmans stain until the stain color in fruits changed. Ascorbic acid content was expressed as mg per 100 g of fruit fresh weight [35]. The sugar content was measured using the reaction of potassium ferricyanide reduction. The procedure is based on the oxidation of carbohydrates to reduce potassium ferricyanide, which, in turn, reacts with iron (III) sulfate producing blue staining. Staining intensity was determined by photocolorimetry using a spectrophotometer AAC (Spectr-1) [28,35]. The sugar content was calculated using optical density, according to the calibration curve. For determination of the total sugar content after hydrolysis, 1 mL of 5% hydrochloric acid was added to test tubes containing 2 mL of the extract and heated for 5 min at 70 °C. The tubes were cooled and neutralized with 5% alkali.

### 2.5. Statistical Analysis of Data

Data are presented as mean values from three-year experimental data with standard deviations. Viability was assessed in a one-way analysis of covariance. Viability data obtained in different treatments were compared and assessed using dependent samples *t*-test. Statistical analysis was performed using Statistica 13 software [36].

## 3. Results

### 3.1. Viability of Cuttings after Cold Storage and Cryopreservation

Baseline (initial) viability of bird cherry cuttings was compared to that after cold and cryogenic storage in LN vapor (Table 2). 

A one-way analysis of covariance revealed no significant difference between the five varieties within the same storage and recovery conditions (*p* = 0.193–0.802). A two-way analysis of covariance showed that the variant of the experiment (storage and recovery condition) had the most prominent effect on viability (*p* < 0.0001), while the effects of the variety (*p* = 0.062) and of factor interactions (*p* = 0.921) were insignificant. The variant of the experiment had the greatest influence (83.7%) on viability variation, while the influence of the genotype was established to be 2.6%, the factor interaction influence was 2.0%, and error was 11.8%. 

The results demonstrated high viability of control cuttings (baseline viability) varying from 86.7% (Avgustina variety) to 93.3% (Sakhalinskaya ustoichivaya variety). Both cold storage and cryopreservation led to a significant decrease in viability compared to the baseline. The viability of cuttings recovered in the laboratory varied from 73.3% to 86.7% after cold storage and from 46.7% to 60.0% after cryopreservation. When recovered in the field, cuttings showed viability within 53.3–60.0% after cold storage and within 43.3–50.0% following cryopreservation. On average, the cryopreservation effect was more detrimental, resulting in a lower mean viability (47.3% and 54.0%) compared to the cold storage (58% and 80%).

The viability of the cuttings planted in the field after cold storage (58.0% mean viability) was significantly lower than in the laboratory (80.0%). By contrast, viability of cryopreserved cuttings remained almost unaffected by recovery conditions (mean viability 47.3% in the field vs. 54.0% in the laboratory). It is important to note that viability of all varieties after cryopreservation for six months exceeded the existing genebank standard of 40% [4,5].

During recovery in the laboratory, cuttings of the Avgustina variety demonstrated the highest viability (86.7%) after cold storage. The Granatovaya grozd’ variety showed high viability both after cold storage (83.3%) and after cryopreservation (60.0%). Avgustina and Sakhalinskaya ustoichivaya demonstrated the best viability (50.0%) after cryopreservation when recovered in the field. 

It is noteworthy that the cold storage and cryopreservation experiments were performed during three consecutive years, and no significant differences in viability were observed between variants of the experiment upon per-year analysis (*p* = 0.065–0.774).

Thus, bird cherry cuttings preserved their viability over 40% after cold and cryogenic storage. The viability of the cuttings planted in the field following cryogenic storage, complied with the standards established for the genebanks and varied from 43.3 to 50.0%.

### 3.2. Morphometric Parameters of the Rooted Cuttings

In addition to viability, morphometric parameters of the rooted bird cherry cuttings were assessed in two variants of the experiment: following cold storage and following cryogenic storage (Table 3).

Rooted cuttings of Avgustina and Sakhalinskaya ustoichivaya varieties are shown in Figure 1 and Figure 2, respectively. The assessed morphometric parameters included plant height, the number of shoots and internodes, and the number and length of roots, as summarized in Table 3.

Plant height for all accessions, except Ranyaya kruglaya, was greater after cold storage compared to cryopreservation (Table 3). The highest number of shoots was recorded after cold storage for the varieties Avgustina (1.3) and Rannaya kruglaya (3.5). After cryopreservation, the number of shoots was highest in the Avgustina variety (1.5). In Rannyaya kruglaya, the number of shoots was significantly reduced after cryopreservation compared to cold storage. In the cold storage group, the highest number of internodes was recorded for varieties of Granatovaya grozd’ (9.5) and Rannyaya kruglaya (15.0). A significantly lower number of internodes was recorded in varieties of Granatovaya grozd’ and Rannyaya kruglaya after cryopreservation compared to cold storage. Root number remained similar between cold and cryogenically stored cuttings in all varieties, while root length was greatly reduced after cryopreservation in all genotypes tested.

### 3.3. Biochemical Composition of Fruits

Upon beginning of the fruiting period, the biochemical composition of bird cherry fruits was analyzed. Table 4 summarizes the results of fruit biochemical analysis in plants grown in the field after cold storage and after cryopreservation. The data were collected over a period of three years, from 2017 to 2019.

No significant difference in dry matter content was observed in bird cherry fruits between the cold storage and cryopreservation groups. The content of dry matter varied within a narrow range of 27.1–29.5%. Similarly, cryopreservation had no significant effect on total sugar content and the concentration of ascorbic acid in fruits of the developed plants. The difference between trees developed after cold storage and cryopreservation did not exceed 1–2%.

Among the varieties, higher levels of sugars were observed in fruits of Samoplodnaya (16.3–16.6%) and Sakhalinskaya ustoichivaya (15.9–16.0%). The level of ascorbic acid was highest in fruits of varieties Granatovaya grozd’ and Samoplodnaya (20–21 mg/100 g fresh weight).

## 4. Discussion

The results of our study demonstrated that cuttings of bird cherry with dormant buds remain viable after dehydration and cryopreservation in LN vapor. These results are in line with the work by Stusnnoff [37] who presented the classification of fruit crops based on their frost resistance and potential for cryopreservation. According to this classification, Class I includes species that are extremely frost-resistant, e.g., *Amelanchier alnifolia* Nutt. And *Prunus virginiana* L. (tolerate temperatures below −60 °C) that demonstrate rapid acclimation, are resistant to dehydration-induced damage, and withstand cryopreservation without prior dehydration. Class II includes frost-resistant species, such as *Malus baccata* Borkh., *P. besseyi* Bailey, *P. pennsylvanica* L.f., and *P. tenella* Batsch, that are also capable of rapid acclimation but are susceptible to dehydration-induced damage; these species also tolerate cryopreservation without prior dehydration. Class III includes the frost-resistant species *M. pumila* Mill., *P. nigra* Ait., *P. fruticosa* Pall., *P. tomentosa* Thunb., and *P. salicina* Lindl. (which survive at –40 °C) that are susceptible to dehydration-induced damage and require controlled acclimation for storage. Class IV includes the cold-susceptible species *P. armeniaca, P. avium* L., *P. cerasus* L., and *P. persica* (L.) Batsch (which tolerate temperatures above −40 °C) with slow acclimation, are susceptible to dehydration-induced damage, that do not withstand direct cryopreservation in LN and require controlled acclimation to recover from cryopreservation. Class V includes highly cold-susceptible species (threshold temperature above −20 °C). Several authors also investigated cryopreservation in fruit crops such as apples, pears, and almonds; their studies demonstrated viability of LN-stored cuttings and buds varying from 50.0% to 70.0% [22,38,39]. For example, in the recent study by Höfer and Flachowsky [40], 180 accessions belonging to 32 species of *Malus* were tested for cryopreservation using the dormant bud method within ten years, with an average recovery rate of 39%. Among those, 116 accessions achieved the 40% threshold recovery.

In our study, both cold storage and cryopreservation led to reduced viability of winter cuttings compared to the baseline viability. We hypothesized that such moderate viability decrease may be caused by the combinational effect of dehydration and low temperatures. On the other hand, statistical analysis revealed only a minor change in viability for cuttings recovered in the laboratory after cold storage, which confirms that the cold storage method routinely used in the VIR genebank is suitable for mid-term conservation of the bird cherry. It is interesting that recovery conditions (in the laboratory or in the field) affected viability of cuttings after both cold storage (significantly) and cryopreservation (insignificantly). This difference may be caused by various factors including more stable physical conditions in the laboratory compared to field. Further experiments are required to better understand the effects of those factors on viability. It is, however, important that, in all varieties tested, viability after cryopreservation was above the standard threshold level of 40% accepted for the genebanks.

The analysis of morphometric parameters of bird cherry cuttings allows for the assessment of growth and development of the plants following storage at low temperature and cryopreservation. A two-way analysis of covariance demonstrated significant difference between varieties in all assessed parameters, except for root length. At the same time, only a few significant variations were noted in plant height, number of shoots, internodes and roots, and root length between cold-stored and cryopreserved cuttings. Other studies [41,42] confirmed the stability of genetic and morphological parameters in various plant materials after cryopreservation. According to Kaity et al. [43], genetic changes may occur in papaya plants following cryopreservation. In general, it is acknowledged that the use of dormant buds for cryopreservation without establishing the in vitro culture reduces the risk of somaclonal variations in plant materials [27] and thus is preferable for the long-term storage of vegetatively propagated fruit trees. 

Biochemical analysis of bird cherry fruits confirmed high levels of dry matter and total sugars and low levels of ascorbic acid in the fruits. No significant difference between the varieties was revealed. Cold storage and cryopreservation had no significant effect on fruit quality in all accessions tested. In our previous works we noted that blackcurrant berries are less susceptible to the effects of cryopreservation, as assessed by mean berry mass, berry mass loss and levels of solids, organic acids, and vitamin C [30]. In gooseberry plants grown after cryopreservation, an increase in mean berry mass, levels of solids, vitamin C and carbohydrates, and a decrease in the levels of organic acids was observed [30].

## 5. Conclusions

Our results confirmed the high reliability of the cold storage method for mid-term conservation of winter cuttings of the bird cherry. The study also demonstrated that cryopreservation of dehydrated winter cuttings in LN vapor could be successfully applied to genotypes of different genetic origin. The viability of the cuttings after cryopreservation determined under laboratory conditions or in the field were similar among genotypes and exceeded the genebank threshold of 40% in all five varieties tested. Cryopreservation had no or minor influence on morphometric parameters (plant height, number of shoots, internodes, and roots) of the recovered plants. Moreover, we found no change in biochemical composition of fruits produced by plants developed from cryopreserved cuttings. In the future, we plan to implement both storage methods to safely backup genetic collections of the bird cherry at the VIR genebank. Cold conditions will be used for mid-term storage and for storing materials for research purposes, while cryopreservation will be used as an ultimate long-term storage back-up of the core collection. Further investigations should involve a wider range of bird cherry species and varieties and include interspecific hybrids, in order to investigate the response of a broader range of genotypes to cryogenic conditions.

## Figures and Tables

**Figure 1 biology-12-01071-f001:**
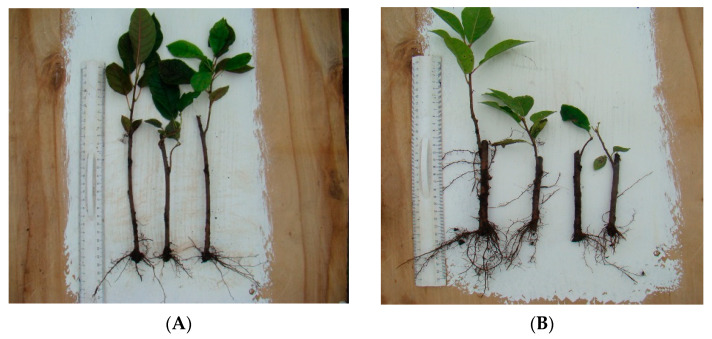
Rooted cuttings of the Avgustina variety: (**A**) after cold storage, and (**B**) after cryopreservation.

**Figure 2 biology-12-01071-f002:**
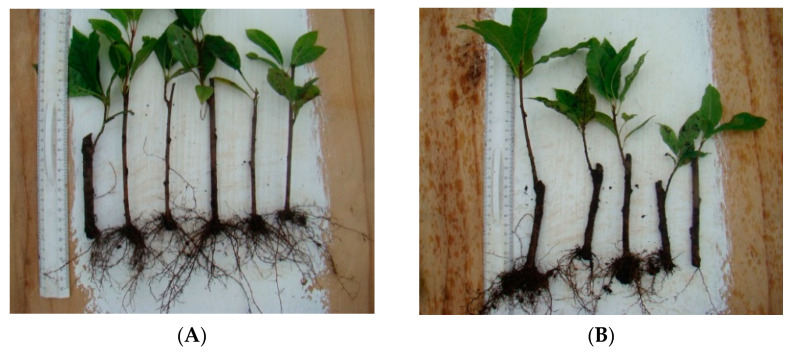
Rooted cuttings of the Sakhalinskaya ustojchivaya variety: (**A**) after cold storage, and (**B**) after cryopreservation.

**Table 1 biology-12-01071-t001:** Bird cherry varieties used in the study.

Variety	VIR Catalog №	Genetic Origin
Avgustina	42,101	*P. virginiana* × *P. avium*
Granatovaya grozd’	42,102	*P. virginiana* × *P. avium*
Rannyaya kruglaya	42,109	Seedling of Pamyati Salamatova (*P. virginiana* × *P. avium*)
Samoplodnaya	42,110	Seedling of Pamyati Salamatova (*P. virginiana* × *P. avium*)
Sakhalinskaya ustojchivaya	42,287	*Padus avium* Mill.

**Table 2 biology-12-01071-t002:** Viability of cuttings of five bird cherry varieties after 6 months of cold or cryogenic storage.

Variety	Viability of Cuttings, %				
	Baseline (Initial) Viability (Laboratory Conditions)	Viability Under Laboratory Conditions		Viability in the Field	
		Cold Storage	Cryopreservation	Cold Storage	Cryopreservation
Avgustina	86.7 ± 3.3 ^fg^	86.7 ± 3.3 ^fg^	56.7 ± 3.3 ^abcd^	56.7 ± 3.3 ^abcd^	50.0 ± 5.8 ^abc^
Granatovaya grozd’	90.0 ± 5.8 ^g^	83.3 ± 3.3 ^efg^	60.0 ± 5.8 ^abcde^	63.3 ± 3.3 ^abcdef^	46.7 ± 3.3 ^a^
Rannyaya kruglaya	86.7 ± 3.3 ^fg^	73.3 ± 3.3 ^bcdefg^	50.0 ± 5.8 ^ab^	53.3 ± 3.3 ^abc^	46.7 ± 3.3 ^a^
Samoplodnaya	90.0 ± 5.8 ^g^	76.7 ± 3.3 ^cdefg^	46.7 ± 3.3 ^a^	56.7 ± 3.3 ^abcd^	43.3 ± 3.3 ^a^
Sakhalinskaya ustojchivaya	93.3 ± 3.3 ^g^	80.0 ± 5.8 ^defg^	56.7 ± 3.3 ^abcd^	60.0 ± 5.8 ^abcde^	50.0 ± 5.8 ^abcd^
Average	89.3 ± 1.8 ^D^	80.0 ± 2.0 ^C^	54.0 ± 2.1 ^AB^	58.0 ± 1.7 ^B^	47.3 ± 1.8 ^A^

The same lowercase letters mark the mean values that do not differ significantly at *p* < 0.05. The same capital letters mark the average values of viability in different treatments that do not differ significantly at *p* < 0.05. Data are mean values from the experiments performed in three consecutive years (200 cuttings per treatment per year) with standard deviations.

**Table 3 biology-12-01071-t003:** Morphometric parameters of rooted bird cherry cuttings after 6 months of cold storage or cryopreservation. Plants developed from cuttings recovered in the field.

Variety	Plant Height, cm		Number of						Length of Roots, cm	
			Shoots		Internodes		Roots			
	Cold	Cryo	Cold	Cryo	Cold	Cryo	Cold	Cryo	Cold	Cryo
Avgustina	25.3 ± 3.3	15.5 ± 0.5	1.3 ± 0.3	1.5 ± 0.5	2.7 ± 0.3	4.5 ± 2.5	10.7 ± 0.9	9.0 ± 0.0	6.7 ± 0.9	10.0 ± 0.0
Granatovaya grozd’	22.5 ± 4.5	19.5 ± 1.5	1.0 ± 0.0	1.0 ± 0.0	9.5 ± 2.5	7.0 ± 5.0	5.5 ± 0.5	4.5 ± 0.5	12.0 ± 2.0	12.0 ± 2.0
Ranyaya kruglaya	8.8 ± 0.3	20.8 ± 4.3	3.5 ± 0.5	1.0 ± 0.0 *	15.0 ± 2.0	3.8 ± 1.0 *	12.5 ± 1.5	12.0 ± 2.4	8.5 ± 2.5	6.8 ± 1.3
Samoplodnaya	12.3 ± 2.2	8.3 ± 0.9	1.0 ± 0.0	1.0 ± 0.0	3.3 ± 0.8	3.0 ± 0.6	8.0 ± 0.7	7.3 ± 1.8	12.3 ± 1.7	8.7 ± 1.3
Sakhalinskaya ustoychivaya	25.5 ± 1.2	22.6 ± 3.3	1.0 ± 0.0	1.0 ± 0.0	2.4 ± 0.4	3.6 ± 1.2	2.2 ± 0.4	8.8 ± 2.4 *	11.2 ± 1.3	9.8 ± 1.7

Cold—cold storage; Cryo—cryopreservation; *—values are significantly different between cold and cryopreservation storage according to a Student’s *t*-test at a 5% significance level.

**Table 4 biology-12-01071-t004:** Biochemical parameters of bird cherry fruits (average from three-year experiments, 2017–2019). Plants developed from cuttings recovered in the field.

Variety	Dry Matter Content (%)		Total Sugars Content (%)		Ascorbic Acid Content (mg/100 g Fresh Weight)	
	Cold Storage	Cryopreservation	Cold Storage	Cryopreservation	Cold Storage	Cryopreservation
Avgustina	27.6 ± 2.1	27.1 ± 2.0	12.1 ± 1.5	13.1 ± 0.5	20.5 ± 1.1	19.4 ± 2.0
Granatovaya grozd’	28.1 ± 2.1	27.0 ± 2.8	14.8 ± 2.1	13.4 ± 3.3	21.1 ± 2.3	20.6 ± 3.1
Rannyaya kruglaya	27.9 ± 1.5	28.0 ± 1.7	15.7 ± 2.0	14.5 ± 3.1	19.8 ± 2.2	19.9 ± 2.8
Samoplodnaya	27.3 ± 3.1	28.1 ± 2.0	16.3 ± 2.1	16.6 ± 1.8	21.1 ± 3.3	21.1 ± 2.9
Sakhalinskaya ustojchivaya	28.1 ± 2.5	29.5 ± 1.1	15.9 ± 2.5	16.0 ± 2.4	19.1 ± 2.6	18.8 ± 3.0
Average	27.8 ± 2.3	27.9 ± 1.9	14.9 ± 2.0	14.7 ± 2.2	20.3 ± 2.3	19.9 ± 2.7

No significant differences between treatments or varieties were recorded.

## Data Availability

The data presented in this study are available on request from the corresponding author. The data are not publicly available due to privacy or ethical restrictions.

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
