# Peer review of "Conservation of the Bird Cherry (Padus Mill.) Germplasm by Cold Storage and Cryopreservation of Winter Cuttings"

_biology, 2023, doi:10.3390/biology12081071_

Round 1
Reviewer 1 Report
All chapters need to be structured more clearly.
Introduction
Why are the control variants described in the last section of the introduction?
Methods
The methods section needs to be structured more clearly and described better. For example, the experiments or variants should be numbered and this numbering should also be used in the text. What were the light space conditions? How many fruits were used for the analyses?
Results
Where are the results on the influence of the year?
What does 3 sm mean in the table?
In the text Rannyaya kruglaya is mentioned with 15.0+/- 2.0, but this is not in any table? What does other indicator mean?
All parameters examined showed significant differences except for the length of the roots? What about the plant height, however, no significance is given here either.
In the second last paragraph on page 6, the table reference is missing.
Figure 3 talks about experiment or experience? Which experiment?
Why are the results repeated in the last paragraph on page 8 and at the beginning of page 9?
Author Response
ANSWER
WE thank the reviewer for all useful comments and advice.
All chapters were structured and revised, The control and experimental variants were described in M M section
The methods section was revised and all and conditions, measuring parameters and number and quantity of used material are indicated.
We have dicided to exclude the results for influence of year because the LSD is not significant
“….In the text Rannyaya kruglaya is mentioned with 15.0+/- 2.0…” - It was a typo, the number 1 out of 15 was accidentally erased (Table 3, revised)
“….All parameters examined showed significant differences except for the length of the roots? What about the plant height, however, no significance is given here either”.
The table 3 shows a significant difference in plant height between the options and this is confirmed in figures 1 and 2.
Figure 3 talks about Content of soluble dry substances in bird cherry fruits.
“Experiment” is right. The experiment – mean the parameters data after cryoconservation od cuttings
Reviewer 2 Report
General Comments
The manuscript compares how four hybrids Padus virginiana x Padus avium and one accession of Padus avium survive cryo-conservation using liquid nitrogen. Three assessments were conducted: (1) the viability of thee stored material was compared to material stored under standard conditions; (2) the rooted bird cherry cuttings were compared using morphological assessments; and (3) assessments of the fruits from stored material in comparison to the original trees were conducted.
The most relevant experiment is the comparison of the viability of the material exposed to cryopreservation with the viability of material stored under standard conditions.
The two other assessments are also relevant, because they show that there is no negative impact of cryo conservation on morphological traits or on chemical fruit characteristics, but are not well presented.
The comparison for morphometric characteristics is not described in the Material and Methods section. That must be added. The statistical stringency of these observations is not evident from the text. No numbers of plants of fruits assessed are reported. It seems these observations are also from different years and one expects major environmental impact. Perhaps these sections need to be reported as anecdotal additions only and cannot have the same bearing as the main experiment. To me the significance of any morphometric changes being associated with cryo-preservation seem not to be evident, although the figures 3 to 5 show on top of the columns small bars pointing at LSDs, but that is not explained by the authors and the quality of the print is too poor to use these bars for comparison.
What is presented in Figures 3, 4 and 5 also does not show any significant differences. These figures can be omitted, in particular if there is no stringency in how the data was generated. There seem to bars on top of the columns.
In Table 2, a row could be added at the bottom to show the overall mean values. This would help to support what is discussed in the second paragraph on the same page below Table 2. It made me suspicious to see that in table 2 many values and the reported standard error were identical for the various accessions but also across different treatments. How is that possible? It made me wondering how the underlying date was generated. If in deed the presented values are mean values based on 200 single observations, one would not expect to see so many identical numbers. The authors must explain this observation in the text.
Specific comments
Abstract: You must explain that you refer to a Russian state standards for viability. Later in the text you must provide a reference to a document for that. You say the vapour phase of liquid nitrogen has a temperature of -183 C. Is that in deed the case in the tanks used? Literature reports sometimes higher temperatures for the vapour phase.
Page 2 lines 1: replace “genetic banks” by “gene banks”.
P. 2, line 9: delete “well known” and delete “of” in the next line.
P.2 line 18: Delete “an increased”.
P. 2 lines 20 to end of paragraph: What starts with “in the control variant….” Does belong into the Material and Methods section.
P. 2, Material and methods, first line: change wording to: ”The research material were 5 varieties….. “
Page 3, first line: Delete “To carry…..cryopreservation”.
P. 3, line 6: Is “thermostat” the right word? Do you mean an “incubator”?
P 3l line 8: Is “germinating” the right term? These are not seeds. Is “sprouting” better?
P 3 line 13: Insert “were” after “incubated”.
P. 3, bullet point no. 5: Replace “Experience” with “Experiment”
P. 3, heading “Experiment Options” What does that mean?
P. 3 in section 2.3, line 3: Replace “according to the methods used for biochemical parameters” by “as follows:”. The entire description is very unclear and needs to be cleared up. From which year were the fruits; how many; any statistical stringency?
Page 4, bottom paragraph: This could be supported by an additional row in Table 2 showing the mean values for each column.
Table 3: Replace “sm” by “cm”.
Page 5: This was note described in Material and Methods.
Page 9, second paragraph: Thus, as a result…” this could replace all text before. You elaborate the analysis much to much given that it is so poorly described.
Page 9, section 4. Discussion, second paragraph, line 2 : What do the kidneys do here?
Author Response
WE thank the reviewer for all useful comments and advice.
The main goal of our research was to prove that cryopreservation does not significantly affect not only the viability of bird cherry cuttings, but also the development of its plants - growth, development of cuttings, as well as the biochemical content of fruits, as indicated by tables and figures that we consider relevant. in the article
In table 2, the same indicators between varieties in response to cryopreservation indicate the absence of influence of the variety on the indicators, and in our opinion this is quite possible, despite the quite similar figures. We allow it
For - Specific comments:
In accordance with all the comments and recommendations, the sections and the text were edited and we tried to correct all errors.
Reviewer 3 Report
1. Abstract, Introduction and Discussion sections should be revised completely (Please see my comments in the text).
2. Cold preservation is not control. It must be a treatment, in my opinion.
3. Keywords must be changed.
4. There are many typo mistakes. Please revised again.
5. Quality of Figures 3-5 is low.

Author Response
WE thank the reviewer for all useful comments and advice.
*. Abstract, Introduction and Discussion sections were revised
* Keywords were changed.
* About Cold preservation as control : In the VIR Genebank in one of the main conservation works. Collection samples of fruit trees are cutting cuttings from these field samples and, since there are quite a lot of these cuttings every year, they are stored at a temperature of -5 degrees before they are placed in cryostorage. In the same way, they are stored until the direct laying of the seedling field nursery. Therefore, in our experience, storage at a temperature of -5 degrees is
* Figures 3-5 were changed (improved).
* On peer-review : We tried to correct all the errors that you found and took into account your recommendations on the content of the text and edited it accordingly in all sections.
Round 2
Reviewer 2 Report
The authors improved the Material and Methods Section by at least somehow describing the morphometrical and chemical analysis. However, these two aspects are not presented and analyzed according to scientific standards. The statistical analysis remains obscure. I advised before to remove the analysis and the figures 3, 4 and 5. Mu suggestion editor requests a third opinion on this matter. I would suggest rejecting the paper based on not correcting these flaws.
Author Response
Thank you fro your suggestions and comments. In the attached file, please open our edited manuscript.
We have edited it in accordance with the your comments and requirements.
Following your recommendation, the figures were removed and replaced with one table showing all the results of biochemical analysis.

Reviewer 3 Report
Good lock
Author Response
Thank you for your comments and suggestions, Thank you again for your time for second review.
Round 3
Reviewer 2 Report
Thank you for you for considering the suggestions made.
Author Response
Thank you again for your time and energy for reviewing this paper